# OpenReview forum: "Advanced Optimization Techniques in Neural Networks: A Sobolev Space Approach"
_TMLR — Rejected by TMLR_

### Review · Reviewer_RgE5 · 2024-11-05

**Summary Of Contributions:**

This paper aims to leverage the derivative information in the Sobolev loss function to obtain better convergence for scientific and engineering applications. Several computational results on the gradients of the Sobolev loss are provided and a preliminary test on simple functions is used to demontrate the effectiveness.

**Audience:**

No

**Claims And Evidence:**

No

**Requested Changes:**

I believe a substantial of the paper needs to be revised to address the weakness points mentioned above.

**Strengths And Weaknesses:**

Weakness:

* There are barely any literature search on related results.
* The idea of using Sobolev norm is by no means new. Similar ideas have been thoroghly investigated in many works, one of which is [1].
* Most of the theorems are well-known results from convex optimization and they don't seem to be related to neural network models. The convex assumption in Theorem 5 does not hold for neural networks. This certainly is against the main goal of the paper.
* Given that the weak part of the theory, a substantial numerical tests are needed. The current numerical section only contains toy examples. It does not even have a comparison to NN trained with MSE loss.


[1] O'Leary-Roseberry, Thomas, et al. "Derivative-informed neural operator: an efficient framework for high-dimensional parametric derivative learning." Journal of Computational Physics 496 (2024): 112555.

---

### Review · Reviewer_qkQC · 2024-11-17

**Summary Of Contributions:**

This paper introduces Sobolev loss for neural network training, extending traditional loss functions by incorporating derivative information. The key contributions include:

Sobolev Loss: The method integrates both function values and derivatives, aiming for smoother and more accurate function approximations.

Theoretical Analysis: The authors provide proofs for gradient descent convergence with Sobolev loss under certain conditions.

Empirical Demonstrations: The approach is tested on simple functions $\sin(x)$ and $e^{-x}$, showing improved approximation and convergence.

**Audience:**

Yes

**Broader Impact Concerns:**

This is a theoretical work. There is no such concern.

**Claims And Evidence:**

Yes

**Requested Changes:**

In the proof of Theorem 3, the authors should mension the Taylor series expansion of $L(\theta)$ with Lagrange remainder, where $H$ is the Hessian matrix of $L(\theta)$ at some point $\theta'$ on the line segment between $\theta_n$ and $\theta_{n+1}$.

Evaluating $L_{\text{Sobolev}}$ numerically seems challenging. The paper should discuss how the points $x_i$ are selected in Algorithm 1: are they chosen deterministically or randomly?

If possible, the paper should also address the challenges that arise when observation errors are present, considering the setting $Y_i = u(X_i) + \varepsilon$.

**Strengths And Weaknesses:**

Strengths:

Theoretical Justification: The mathematical analysis is comprehensive, with clear convergence proofs.

Potential Impact: The approach is applicable to tasks where smooth function approximation is critical, particularly in scientific and engineering applications.

Weaknesses:


Seemingly Limited to Known Function Approximation: The current framework appears tailored for approximating a known target function $u(x)$ within the Sobolev space. This may restrict its applicability to problems where the true function is explicitly defined.

Limited Generalization to Unknown Functions: The approach could be generalized to a broader $L^p(\Omega, \mu)$ setting with a probability measure $\mu$. In practice, we often encounter an interpolation problem where the target function $u: \mathbb{R} \to \mathbb{R}$ is unknown, and we only observe $ Y_i = u(X_i)$ based on i.i.d. samples $X_1, \cdots, X_n $. It is uncommon, however, to have access to observations of the derivative $Z_i = u'(X_i)$, which limits the practical feasibility of the proposed method in real-world scenarios.

---

> ### Comment · Reviewer_qkQC · 2024-12-28
> **Office Comment**
>
> I agree with AC's opinion to reject the paper.

---

### Review · Reviewer_zbiG · 2024-11-28

**Summary Of Contributions:**

The paper provides a compelling discussion on Sobolev loss, which incorporates derivative information to enhance neural network training, addressing the limitations of conventional loss functions like MSE or MAE. This perspective is particularly valuable for applications requiring smooth function approximations.

**Audience:**

Yes

**Broader Impact Concerns:**

I have no such concerns.

**Claims And Evidence:**

Yes

**Requested Changes:**

- (Important) Please rearrange the figures and tables in the paper. Now, the layout needs to be better structured.
- (Important) Discuss computational overhead and potential methods (e.g., efficient automatic differentiation) to mitigate it. Benchmark Sobolev loss against other derivative-aware loss functions or training methods.
- Explore strategies for scaling Sobolev loss to high-dimensional tasks, possibly by leveraging sparsity or GPU acceleration.

**Strengths And Weaknesses:**

Strengths:
- **Theoretical Foundations**: The use of Sobolev spaces and norms is rigorously introduced with clear definitions and proofs of related theorems, such as gradient computation and convergence.
- **Convergence Analysis**: The paper extends classical optimization results to Sobolev loss, proving convergence under certain conditions, which enhances its theoretical robustness.
- **Clarity of Computational Examples**: The empirical results, particularly the detailed loss progression tables and function approximations, effectively illustrate the benefits of Sobolev loss.

Weaknesses:
- **Limited Scope of Applications**: While the examples of $\sin(x)$ and $e^{-x}$ effectively demonstrate the technique, these are relatively simple and well-behaved functions. The paper would benefit from testing on more complex, real-world scenarios.
- **Generalization Discussion**: The paper asserts improved generalization with Sobolev loss but lacks rigorous testing across diverse datasets or architectures to substantiate this claim.
- **Practical Challenges**: The scalability of Sobolev loss, particularly for complex tasks with high-dimensional inputs and outputs, is not addressed.

---

> ### Comment · Reviewer_zbiG · 2024-12-28
> **Office Comment**
>
> I agree with AC's opinion.

---

### Decision · Action_Editor_Wrwu · 2025-01-21

**Recommendation:** Reject

**Comment:**

This paper advocates the use of the Sobolev loss in machine learning to promote smoother function approximations. To this end, it presents a theoretical analysis of gradient descent with the Sobolev loss, alongside simple numerical examples. While this suffices for a simple expository paper and the reviewers appreciate its clarity in general, all reviewers find the paper inadequate for publication at TMLR. Here is a summary of the main weaknesses:

- The Sobolev loss has already been extensively studied in the literature, but the paper does not sufficiently survey such related works.

- The theoretical analysis in this paper follows the standard facts in convex optimization theory, and fails to provide substantial insights for machine learning applications.

- The experiments only contain toy examples outside the scope of machine learning, and reasonable baselines are missing.

Besides, the authors did not respond to the reviewers' comments.

Based on the above I would like to recommend rejection.

**Audience:**

As a general exposition, the paper could be interesting to certain members of the community. But as it stands, the originality of the paper is substantially lacking.

**Claims And Evidence:**

The paper claims the advantages of the Sobolev loss for machine learning tasks, but does not provide sufficient theoretical and empirical evidences to support this claim. In particular, the experiments in the paper are overly simplified, and lacks sufficient comparison to basic baselines.